# Research on Database Construction and Calculation of Building Carbon Emissions Based on BIM General Data Framework

Ruizhe Zhang , Hong Zhang *, Shangang Hei and Hongyu Ye

Institute of Building Technology and Sciences, School of Architecture, Southeast University,
Nanjing 211189, China; 230169003@seu.edu.cn (R.Z.); 230198409@seu.edu.cn (S.H.); 230218003@seu.edu.cn (H.Y.)
* Correspondence: zhangh555@aliyun.com

**Abstract:** China is entering a new era characterized by carbon peaking and carbon neutrality, and the construction industry, which accounts for a high proportion of social carbon emissions, urgently needs a method to calculate and predict building carbon emissions in advance. This study proposes a method for calculating the life cycle carbon emissions (LCCEs) of buildings based on building information modeling (BIM) technology. The method uses a BIM universal data framework to establish a building carbon emission calculation model and a building carbon emission factor database instance. Taking prefabricated construction projects as an example, it is compared with the traditional calculation method. The results show that the method can more accurately predict building carbon emissions and provide methods and a basis for the construction industry to control carbon emissions in advance.

**Keywords:** prefabricated building; BIM; IFC; LCCE; automatic calculation

## 1. Introduction

Climate change is a global issue of concern today. As the largest developing country in the world, many cities in China are in a stage of rapid urbanization and industrialization [1], and the national carbon emissions account for 33% of the world's carbon emissions (2021) [2], while according to the "2022 China Building Energy Consumption and Carbon Emission Research Report" statistics [2], the total carbon emissions from building and construction account for 50.9% of the national carbon emissions. In order to achieve China's promise of peaking carbon emissions by 2030 and achieving carbon neutrality by 2060 [3], it is necessary to control and reduce carbon emissions from the building industry. There are various ways and technical means to reduce carbon emissions from buildings, but to accurately grasp the impact of each link in the building industry chain on the overall carbon emissions, it is necessary to improve the accuracy and speed of building carbon emission calculation technology, so that the impact variables of building carbon emissions can be more intuitively and accurately displayed in the planning and design stage, in order to make decisions and adjustments.

To calculate the carbon emissions of the whole life cycle of a building, the life cycle of a building should be defined first. Opinions diverge greatly on the division of building life cycle. For example, Liu Boyu [4] (construction, use, and demolition) and She Jieqing [5] (materialization, use, and demolition) support the division of building life cycle into three stages. Some believe that the life cycle of a building should be divided to four stages, including Leif and Cole [6] (raw material production, building construction, building use, and building demolition and material disposal), Dong Lei [7] (building material production, construction, use and maintenance, and building demolition), Li Jing [8] (design, materialization, use and maintenance, and demolition and recycling), Gerilla [9] (material production, building construction, building maintenance, and building use) and Bribian [10] (production, construction, use, and end). Those that support the five-stage division are Yu Ping et al. [11] (raw material production, building construction, building

use, maintenance, and disposal). After comparing the differences between traditional construction methods and prefabricated building assembly methods, Wang Yu [12] from Southeast University concluded on six stages of the whole life cycle of industrial prefabricated buildings: building material exploitation, component factory production, logistics, assembly, use and maintenance, and demolition and recycling. Existing research does not have a very unified definition of the division of the building life cycle but is more based on the needs of scholars in their respective research fields on the basis of following objective reality.

In terms of algorithms, the calculation methods of building carbon emissions can be roughly divided into input–output analysis methods and process analysis methods. The input–output analysis method uses the input–output table for calculation from a macro perspective. For example, Nässén [13] and Han [14] proposed a method of using the input–output model to estimate the carbon emissions in the building construction stage and conducted a case study. Acquaye et al. [15] used Monte Carlo simulation to analyze the solidified carbon emissions of apartment buildings on the basis of input–output analysis. Nevertheless, the input–output analysis method is relatively rough and could not optimize the carbon emission sources of buildings. By contrast, the process analysis method focuses on processes of a building's life cycle. For example, Harmouche et al. [16] developed a building construction process analysis program for carbon emission estimation and process identification using data provided by material suppliers. Abanda et al. [17] conducted a comprehensive analysis on the mathematical model of carbon emission quantification in construction projects. Moon et al. [18] reported through case comparison that the error of estimating building carbon emissions in the design stage by process analysis methods was within 8%, which had good accuracy.

With the rapid popularization of building information modeling (BIM) programs, BIM technology has been applied to the management and analysis of carbon emissions in the whole life cycle of buildings. Plebankiewicz et al. [19] considered the ability to automatically prepare the bill of quantities one of the key advantages of BIM. Eleftheriadis et al. [20] combined life cycle assessment (LCA) and BIM to study the current situation of energy conservation in building structural systems and believed that the integration of BIM and LCA could realize the automatic extraction of material quantities.

In terms of regression analysis and formula fitting of buildings, Frame [21] gave the estimation formula of carbon emissions from heating, cooling, and lighting of buildings and compiled a calculation program through Excel. In addition, Luo et al. [22] put forward a regression formula for carbon emissions in the construction process based on the number of floors and the amount of reinforced concrete through the analysis of 78 office buildings in China.

Studies have also been carried out on calculation of the life cycle carbon emission (LCCE) of buildings through LCA. For example, Li et al. [23] proposed the method and steps for calculating carbon emissions in the construction stage based on BIM technology. Peng [24] used Ecotect to build a BIM and, based on this, proposed a method for calculating LCCE of buildings and the calculation boundaries and limitations of this method. After establishing the BIM through Revit, Stadel et al. [25] used IESVE and SimaPro to conduct a case study on the carbon emissions of office buildings and discussed the impact of BIM parameter adjustment on the LCCE of buildings. Ajayi et al. [26] used ATHENA Impact Estimator to calculate the potential value of greenhouse gas emission based on BIM, aiming to evaluate the impact of building materials on the environmental performance during the whole life cycle of buildings. Gardezi et al. [27] adopted the database Inventory Carbon and Energy (ICE) to calculate the LCCE of residential buildings in Malaysia so as to predict the carbon emissions of residential buildings. However, the current BIM-based carbon emission research has not yet involved the calculation of the carbon emissions of the building lifetime at the beginning of the scheme and the optimization design of the building plan using the carbon emission estimation data.

In summary, the existing research, whether it is carbon emission calculation methods or carbon emission calculation tools, lacks the ability to intuitively provide architects with the ability to predict the carbon emissions of the whole life cycle of the building plan at the scheme stage, nor can it provide the impact weight of various types of carbon emissions at the design stage, so as to provide data support for scientifically reducing the total carbon emissions of the project. Based on the process analysis method, this paper comprehensively simplifies the existing building carbon emission regression formula into three types of formulae: human, machine, and material, and it intends to provide this part of the content based on the BIM automatic calculation method of building carbon emissions and provide a theoretical basis for the construction of an automated platform integrating modeling, calculation, and optimization of building modeling.

## 2. Methodology

### 2.1. System Boundary

After learning about and summarizing the European standard (BS EN15978:2011) [28], the Chinese standard (GBT 51366-2019) [29], and the existing research, it can be considered that the mainstream academic view divides the building life cycle into four stages: production stage, construction stage, use stage, and end of life stage. Based on the requirements of building industrialization in most cities in China and the initiative of building reuse, the authors believe that the building life cycle can be refined and extended to seven stages: material preparation, component production, component transport, component assembly, operation and maintenance, renovation and reuse, and demolition and reuse. The description of the 7 stages, boundary definition, and carbon emission calculation formula will be described in detail below.

(1)　Material Preparation Stage

Carbon emissions of this stage refer to the carbon emissions generated in the process of artificial mining and processing of building materials from original storage in nature to form building components. Taking steel used in construction as an example, the carbon emissions from iron ore mining, transportation, smelting, steel production, and other processes are counted in the material preparation stage, but the carbon emissions from processing of steel into beams or columns are not included in this stage. The boundary of the material preparation stage is before the building material enters the component processing plant or building construction site.

(2)　Component Production Stage

The statistical boundary of carbon emissions at this stage is from the entry of basic materials into component processing plants to the delivery of building components, during which the carbon emissions generated by secondary processing, transportation, and storage of materials are all included in the carbon emissions of this stage. Statistically, the carbon emissions during the production of each building component, including the carbon emissions generated by the personnel, machines, and materials involved in each component during this process, should be counted in the carbon emissions of components.

(3)　Component Transport Stage

This stage refers to the process in which building components are loaded, transported to the construction site from the component production plant, and stored at the component yard. If the components are in the stage of demolition and reuse, the carbon emissions generated in the process of loading and transporting the components from the original construction site to the current construction site and storage are counted. It should be noted that the carbon emissions generated by the personnel involved in the loading and unloading of components and the equipment (active) used should be included in the carbon emissions of this stage.

(4) Component Assembly Stage

The carbon emissions of this stage are the carbon emissions generated in the whole process of assembling various building components at the construction site into buildings, including the carbon emissions generated by the personnel, machines, and materials involved in all components in this process. Its boundary is the interval from the construction site to the start of building construction to the completion of building acceptance.

(5) Operation and Maintenance Stage

The carbon emissions generated at this stage are mainly divided into two types, one is the carbon emissions from operation (equipment energy consumption), the other is the carbon emissions from maintenance (component replacement).

Carbon emissions from operation are further divided into energy consumption carbon emissions and energy production carbon emissions. The former is the carbon emission generated by all the energy consumed by the equipment during the operation of the building. A more refined calculation can compare the energy consumption before and after the green and energy-saving optimization design of the building, which will not be discussed in this paper. The latter refers to the negative carbon emissions formed by the energy produced by the energy production equipment during the operation of buildings that use clean energy production equipment.

Carbon emissions from maintenance are the carbon emissions caused by the necessary maintenance and replacement of building components in order to keep the building in normal use during its service life. Different from the renovation and reuse stage, the maintenance carbon emissions only include the carbon emissions generated by the local repair and replacement of components for maintaining their original design function and not those caused by the replacement and upgrading of major structural, maintenance, and equipment components. The carbon emissions of personnel and equipment generated during maintenance are also included in this stage.

(6) Renovation and Reuse Stage

The service life of different types of building components varies. Usually, structural and enclosure components of building have a service life of over 50 years, the service life of equipment components is 20–30 years, and that of decoration components is about 10–15 years. In order to ensure the use function of the building, the building components should be replaced or upgraded when necessary. The change in the owner's demand for the building function will also lead to the replacement and upgrade of components.

In this process, there are two types of renovation and reuse, one is in situ renovation and reuse and the other is ex situ renovation and reuse. In situ renovation and reuse mean that the location of the building itself does not change. By replacing building components or changing the function of the building, the durability of the building can be improved, so that the building function can meet the new demands, thereby extending the service life of the building and reducing the carbon emission intensity. Ex situ renovation and reuse mean to construct a building by reusing the building components from the original building on another site, so as to reduce carbon emissions. The reuse of the original building components whose service life has not reached the limit in the renovation of other buildings is also ex situ renovation and reuse.

(7) Demolition and Reuse Stage

The carbon emissions at this stage are the carbon emissions generated by the process of dismantling and recycling of components after the end of the building life.

*2.2. Carbon Emissions Calculation Method*

2.2.1. Carbon Emission Calculation of Building Life Cycle

The total carbon emission of a building is the sum of the incremental carbon emission of building components in each stage of the building (see Equation (1)). The carbon emission data of each stage of the building are composed of three parts: carbon emissions of

"personnel, machines, and materials", where "materials" include components and supporting materials. Therefore, the carbon emissions of components, personnel, equipment, and supporting materials at each stage of the building should be calculated (see Equation (2)).

$$C_t = C_{mp} + C_{cp} + C_t + C_a + C_{om} + C_u + C_r \tag{1}$$

$C_t$ is the total carbon emission of a building;
$C_{mp}$ is the total carbon emission in the material preparation stage;
$C_{cp}$ is the carbon emission increment in the component production stage;
$C_t$ is the carbon emission increment in the component transport stage;
$C_a$ is the carbon emission increment in the component assembly stage;
$C_{om}$ is the carbon emission increment in the operation and maintenance stage;
$C_u$ is the carbon emission increment in the renovation and reuse stage;
$C_r$ is the carbon emission increment in the reuse stage.

$$C_s = C_c + C_p + C_e + C_{sm} \tag{2}$$

$C_s$ is the carbon emission increment at each stage;
$C_c$ is the carbon emissions of components;
$C_p$ is the carbon emissions of personnel;
$C_e$ is the carbon emissions of equipment;
$C_{sm}$ is the carbon emissions of supporting materials.

For the above four types of carbon emission data, the seven life cycle stages of buildings have their different calculation rules:

### 2.2.2. Carbon Emissions Calculation of Material Preparation Stage

Since the current carbon emission factor of materials already includes indirect and implied carbon emissions from mining, processing, and manufacturing of materials, the carbon emissions at this stage only need to calculate the emissions of all kinds of materials in all components.

$$C_{mp} = \sum_{i=1}^{n} \left( \sum_{a=1}^{n} F_a \times Q_a \right) \times Q_i \tag{3}$$

$I$ is the type $i$ building components;
$Q_i$ is the quantity of type $i$ building components;
$F_a$ is the carbon emission factor of type a materials (carbon emission factors are set according to the standard carbon emission factors published by IPCC. Different countries and regions can also set different values according to the local authoritative carbon emission factor reports);
$Q_a$ is the quantity of type a materials in type $i$ building components.

### 2.2.3. Carbon Emissions Calculation of Component Production, Transport, and Assembly Stages

Although the content of carbon emission calculation in the three stages (component production, transport, and assembly stages) is different, the calculation structure is consistent, so the calculation formula of carbon emissions are explained together. Since the carbon emissions of materials in the components have been included in the material preparation stage, the component will not generate additional carbon emission increments in these three stages. Thus, the component carbon emissions are not included in the calculation.

$$C_{cp/t/a} = \sum_{i=1}^{n} (C_p + C_e + C_{sm}) \tag{4}$$

$C_p$ is the carbon emissions of personnel at this stage;
$C_e$ is the carbon emissions of equipment at this stage;
$C_{sm}$ is the carbon emissions of supporting materials at this stage.

$$C_p = \sum_{a=1}^{n} P_a \times F_p \times T_{p \cdot a} \times Q_a \tag{5}$$

$P_a$ is the number of workers required in process a when processing type $i$ components at this stage;

$F_p$ is the standard time carbon emission factor of personnel, the value of which is the same as that of fuel carbon emission factor);

$T_{p \cdot a}$ is the personnel-hours required for process a when processing type $i$ components at this stage.

$$C_e = \sum_{a=1}^{n} E_a \times F_{e \cdot a} \times T_{e \cdot a} \times Q_a \tag{6}$$

$E_a$ is the energy consumption intensity of the type a equipment used to process type $i$ components at this stage;

$F_{e \cdot a}$ is the energy carbon emission factor of type a equipment, the value of which is the same as that of material carbon emission factor);

$T_{e \cdot a}$ is the running time of type a equipment for processing type $i$ components at this stage.

$$C_{sm} = \sum_{a=1}^{n} M_a \times F_a \times Q_a \tag{7}$$

$M_a$ is the quantity of type a supporting materials used for processing type $i$ components at this stage.

### 2.2.4. Carbon Emissions Calculation of Operation and Maintenance Stage

$$C_{om} = C_o + C_m \tag{8}$$

$C_o$ is the increment of operational carbon emissions;
$C_m$ is the increment of maintain carbon emissions.

$$C_o = (E_c - E_p) \times F_e \times T_o \tag{9}$$

$E_c$ is the energy consumption intensity of the building;
$E_p$ is the energy production intensity of the building (Kwh/y);
$F_e$ is the energy carbon emission factor;
$T_o$ is the operation time of the building (y).

$$C_m = \sum_{i=1}^{n} (C_i + P_i + E_i + SM_i) \times Q_i \tag{10}$$

$C_i$ is the carbon emissions of the type $i$ components to be replaced;
$P_i$ is the carbon emissions of personnel required for replacing type $i$ components;
$E_i$ is the carbon emissions of the equipment required for replacing type $i$ components;
$SM_i$ is the carbon emissions of the supporting materials required for replacing type $i$ components.

$$C_i = \sum_{i=1}^{n} \left( \sum_{a=1}^{n} a \times F_a \times Q_a \right) \times Q_i \tag{11}$$

$Q_i$ is the quantity of type $i$ components;
$a$ is the a-type material constituting type $i$ components;
$F_a$ is the carbon emission factor for type a materials;
$Q_a$ is the quantity of type a materials in type $i$ components.

$$P_i = \sum_{a=1}^{n} P_a \times F_p \times T_{p \cdot a} \tag{12}$$

$P_a$ is the number of workers required for process a when replacing the i-type components;

$F_p$ is the standard time carbon emission factor of personnel;
$T_{p \cdot a}$ is the personnel-hours required in process a for replacing the i-type components.

$$E_i = \sum_{a=1}^{n} E_a \times F_{e \cdot a} \times T_{e \cdot a} \tag{13}$$

$E_a$ is the energy consumption intensity of type $a$ equipment used to replace type $i$ components;

$F_{e\cdot a}$ is the energy carbon emission factor of type $a$ equipment;

$T_{e\cdot a}$ is the running time of type $a$ equipment for replacing type $i$ components.

$$SM_i = \sum_{a=1}^{n} M_a \times F_a \tag{14}$$

$M_a$ is the quantity of type $a$ supporting materials used to replace type $i$ components.

### 2.2.5. Carbon Emissions Calculation of Renovation and Reuse Stage

Calculation formula of carbon emissions from in situ and ex situ renovation and reuse:

$$C_{ul/uo} = C_{il}/C_{io} + \sum_{i=1}^{n}(P_i + E_i + SM_i) \times Q_i \tag{15}$$

$C_{il}$ is the change in carbon emissions due to in situ renovation and reuse (including the carbon emissions generated by renovation and the carbon emissions reduced by the extension of building life due to renovation);

$C_{io}$ is the change in carbon emissions due to ex situ renovation and reuse (same as above);

$P_i$ is the carbon emissions of personnel required for retrofitting type $i$ components;

$E_i$ is the carbon emissions of equipment required for retrofitting type $i$ components;

$SM_i$ is the carbon emissions of supporting materials required for retrofitting and replacing type $i$ components.

$$C_{il} = \sum_{i=1}^{n} C_{im} \times N_u - C_m \times \frac{T_e}{T_d} \tag{16}$$

$C_{im}$ is the total carbon emissions of type $i$ components during materialization (from material preparation stage to component assembly stage);

$N_u$ is the number of renovations of the building;

$C_m$ is the total carbon emissions of the building during materialization (from material preparation stage to component assembly stage);

$T_d$ is the design service life of the building;

$T_e$ is the extended service life of the building due to renovation and reuse.

$$C_{io} = \sum_{i=1}^{n} C_{im} \times N_u - C_m \times N_u \tag{17}$$

$$C_{im} = \sum_{i=1}^{n}(C_{mp\cdot i} + C_{cp\cdot i} + C_{t\cdot i} + C_{a\cdot i}) \times Q_i \tag{18}$$

$C_{mp\cdot i}$ is the carbon emissions of type $i$ components in the material preparation stage;

$C_{cp\cdot i}$ is the carbon emission increment of type $i$ components in the component production stage;

$C_{t\cdot i}$ is the carbon emission increment of type $i$ components in the component transport stage;

$C_{a\cdot i}$ is the carbon emission increment of i-type components in the component assembly stage.

$$C_m = C_{mp} + C_{cp} + C_t + C_a \tag{19}$$

The carbon emission formula of personnel is the same as Equation (12), the carbon emission formula of equipment is the same as Equation (13), and the carbon emission formula of supporting materials is the same as Equation (12).

### 2.2.6. Carbon Emissions Calculation of Demolition and Reuse Stage

$$C_r = C_d + C_t + C_h \tag{20}$$

$C_d$ is the carbon emissions from component disassembly;

$C_t$ is the carbon emissions from transportation of discarded components;

$C_h$ is the carbon emissions from disposal of discarded components.

The carbon emission formulae of $C_d$ and $C_t$ are the same as Equation (4).

$$C_h = C_{lf} + C_{re}$$

$$C_{lf} = \sum_{i=1}^{n}(C_{wt \cdot i} + C_{de \cdot i}) \times Q_i$$

$$C_{re} = \sum_{i=1}^{n}(C_{rep \cdot i} - C_{rec \cdot i}) \times Q_i \tag{21}$$

$C_{lf}$ is the carbon emissions from landfill of discarded components;

$C_{re}$ is the carbon emissions from recycling discarded components;

$C_{wt \cdot i}$ is the carbon emissions from garbage disposal of type $i$ components;

$C_{de \cdot i}$ is the carbon emissions from degradation of type $i$ components after landfill (please refer to the national/regional carbon emission catalogue of garbage disposal for details);

$C_{rep \cdot i}$ is the carbon emissions from reprocessing of type $i$ components;

$C_{rec \cdot i}$ is the carbon emission saved by recycling and processing of type $i$ components (please refer to the national/regional carbon emission catalogue of material recycling for details).

The carbon emission formulae of Cwt·I and Crep·I are the same as Equation (4).

### 2.3. Carbon Emission Data Framework Based on IFC

After defining the carbon emission calculation methods for each stage of the whole life cycle of prefabricated buildings, in order to further improve the statistical efficiency and accuracy of building carbon emission data, it is necessary to introduce BIM tools so that the carbon emission data of each component have a visual and searchable information carrier. At present, there are a variety of BIM tools, each of which has its own weaknesses and strengths. To make the carbon emission calculation of prefabricated buildings available for public use, the framework for carbon emission data of prefabricated buildings should adopt an open-source data structure compatible with most BIM tools and have good scalability. In view of this, our team chose the IFC data structure. Although IFC has some problems as an open-source BIM data structure, its update frequency is stable. In addition to its open source, this framework is also advanced and extensible, and is applicable to most BIM software. Therefore, our team believes that IFC has good prospects among various general BIM structures.

Although the IFC framework defines almost all building components in the architectural field and provides good support for most BIM software on the market [30], its overall structure is designed from a semantic perspective, the attributes and relationships of building elements are modeled in an object-oriented way, and the geometric expression and spatial relationship of building elements are implicitly expressed, without relevant explicit description [31,32]. This leads to the difficulty of dealing with complex geometric relationships and complex data relationships in the model.

### 2.3.1. Brief Description of IFC Data Framework

The IFC standard can describe all aspects of building products and is the most comprehensive and detailed specification for building information. This paper mainly studies IFC2x3 finil, the version most compatible with BIM software at present. This paper will not describe the structure, content, and expansion mechanism of IFC in detail. Instead, it only shows the data relevance of IFC through the description of the entity IfcRoot.

The entity IfcRoot is the abstract base entity of all entities that can independently exchange data. Figure 1 shows its properties and inheritance relationship.

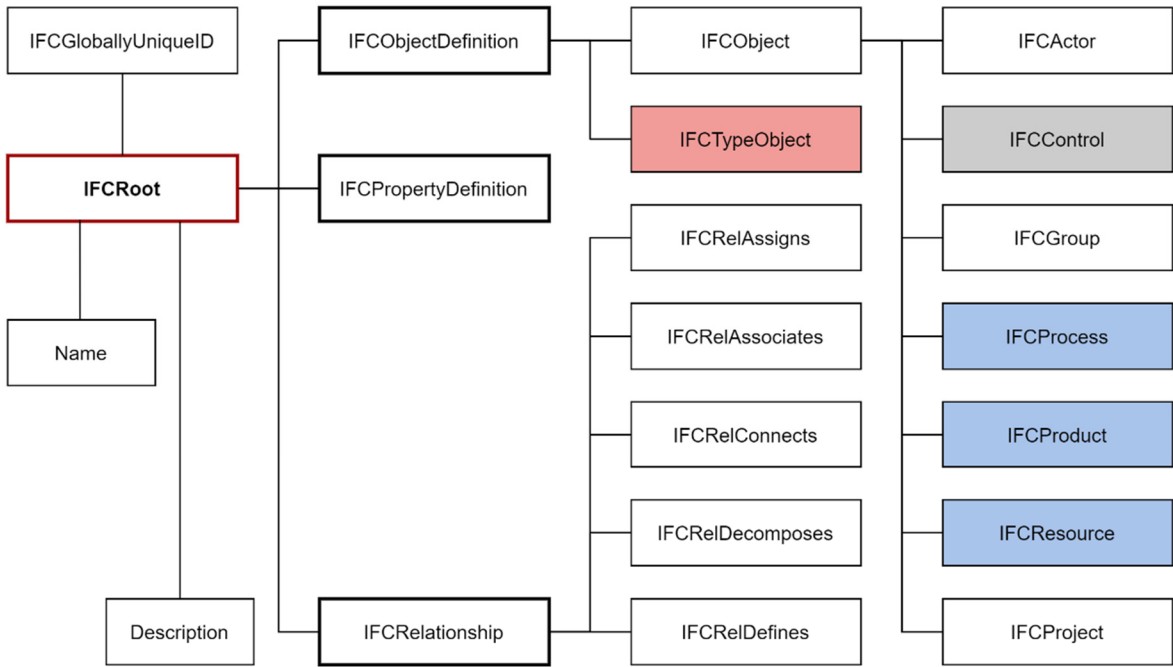

**Figure 1.** Entity IfcRoot and its main inheritance relationships.

2.3.2. Carbon Emission Data Framework

Since there is currently no entity that directly gives carbon emission information in the IFC standard, it is necessary to define the entity of carbon emission information in the IFC framework according to the expansion rules of IFC. However, since building carbon emission information and building cost information are highly similar in content and structure, the self-defined carbon emission information entity can be constructed by referring to the items of the cost information entity IFCCost prefix. Its self-defined carbon emission information will take IFCCarbon as the information prefix.

After expressing the engineering information of the IFC model, a BIM-based carbon emission information framework can be obtained, as shown in Figure 2. This model involves core entities such as IfcProduct (describing building products), IfcProcess (describing progress), IfcResource (describing resources), and IfcControl (describing control) and connects them through relation entities.

In this framework, to represent the relationship between construction tasks and entity components, a one-to-many association between entities IfcProcess and IfcProduct is established through the relationship entity IfcRelAssignsToProcess. To represent resource consumption, entity IfcResource is associated with entity IfcProduct or IfcProcess via the relationship entity IfcRelAssignsToResource. To represent the sub-item information of building product elements, the relationship between IfcTypeObject and entity component is established through the relationship entity IfcRelDefinesByType. One of the attributes of the entity IfcTypeObject is a property set that records the sub-item information. To represent the engineering quantity information of building product elements, the engineering quantity entity IfcElementQuantity is associated with entity components through IfcRelDefinesByProperties. To represent the association between the entity component and the carbon emission project, the relationship between the entities IfcCarbonItem and IfcProduct is established through IfcRelAssignsToControl, so as to realize the carbon emission description of the entire project. In addition, the relationship between the entities IfcWorkSchedule and IfcTask is established through the relationship entity IfcRelAssingsTasks to reflect the distribution of carbon emissions in the entire construction process. Furthermore, the relationship between the entities IfcCarbonSchedule and IfcCarbonItem is established through the relationship entity IfcRelSchedulesCarbonItems to realize the control and management

of carbon emissions. By these means, the distribution of carbon emissions with the progress can be reflected.

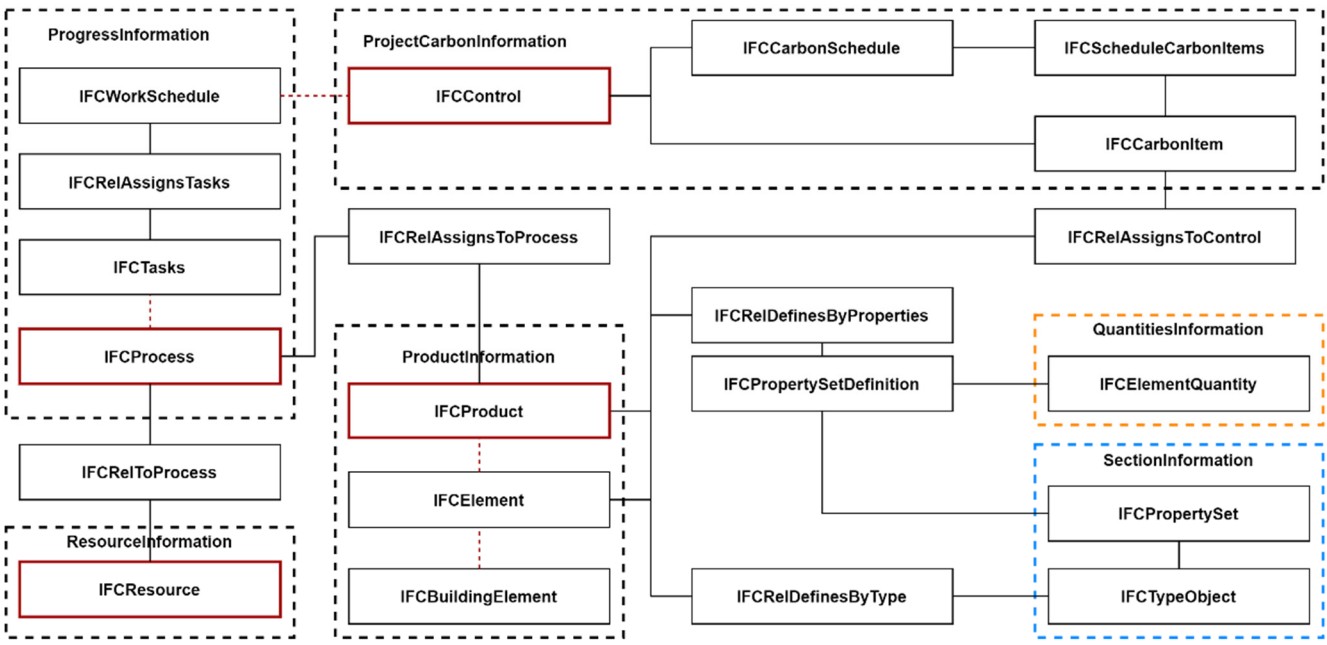

**Figure 2.** Carbon emission data framework.

## 3. Establishing Dynamic Calculation Method Based on Carbon Emission Database

### 3.1. Missing Data of Recording of Real-Time Information of Engineering in IFC Data Structure

Although the attempt to establish a building carbon emission information framework based on IFC is successful, after further analysis of the time parameters of the framework, it is found that the access to the time information cannot be adjusted according to the actual situation. The parameters for time control of the work schedule in the IFC framework are mainly in the IFCWorkSchedule entity under IFCControl and IFCTasks entity under IFCProcess. They realize one-to-many correspondence through the relationship entity IfcRelAssignsTasks, as shown in Figure 3, so as to control the work schedule. This relationship set mainly defines properties about schedule management, rather than specific duration information.

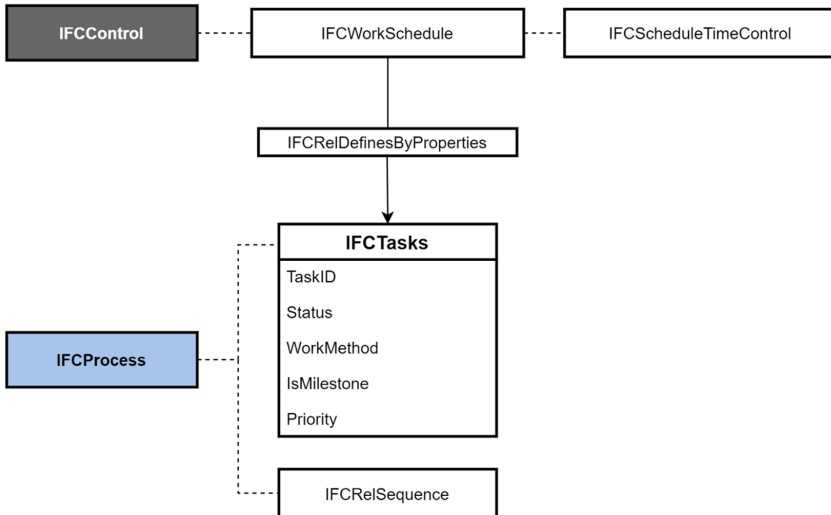

**Figure 3.** Task time control in IFC framework.

The real quantitative indicator of the working time is in the IFCQuantityTime in the IFCPhysicalQuantity under the IFCElementQuantity entity, and the IFCQuantityTime itself is not the working time. The working time is determined by the engineering quantity converted from the physical parameters in the derived entity IFCPhysicalSimpleQuantity through the quota, as shown in Figure 4. In other words, in the IFC framework, the actual working time of each process is not the real source used to define the quantity. The quantity is only related to the physical parameters of the materials involved in the process, and the time is only reflected in the project management. This shows that the data model of the IFC framework is a static model based on the idealized simulation of the quota.

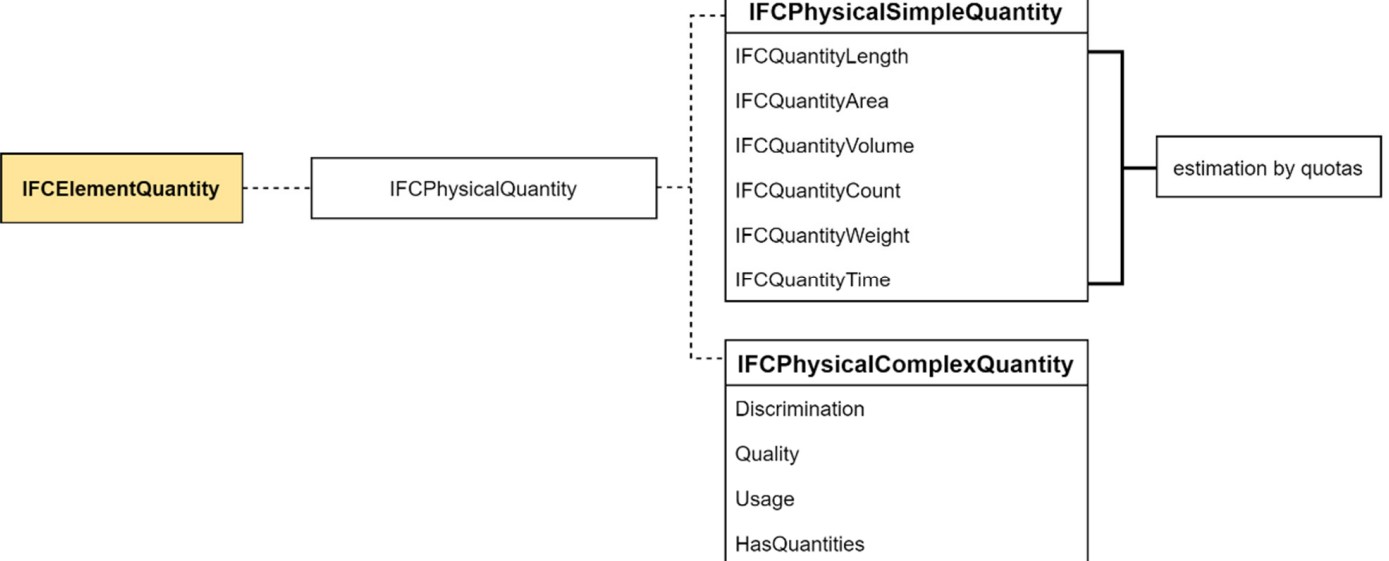

**Figure 4.** Relationship between quantity and working time in the IFC framework.

*3.2. Framework of Automatic Calculation Method Based on Database*

The introduction of a database can solve the problem of dynamic computing under the IFC framework. As shown in Figure 5, the carbon emission data structure under the IFC framework can be integrated into the component information table, process information table, material information table, equipment information table, personnel information table, and carbon emission factor table in the carbon emission database. Then, through the automatic retrieval and operation of database data, the automatic calculation of carbon emission data of prefabricated buildings can be realized.

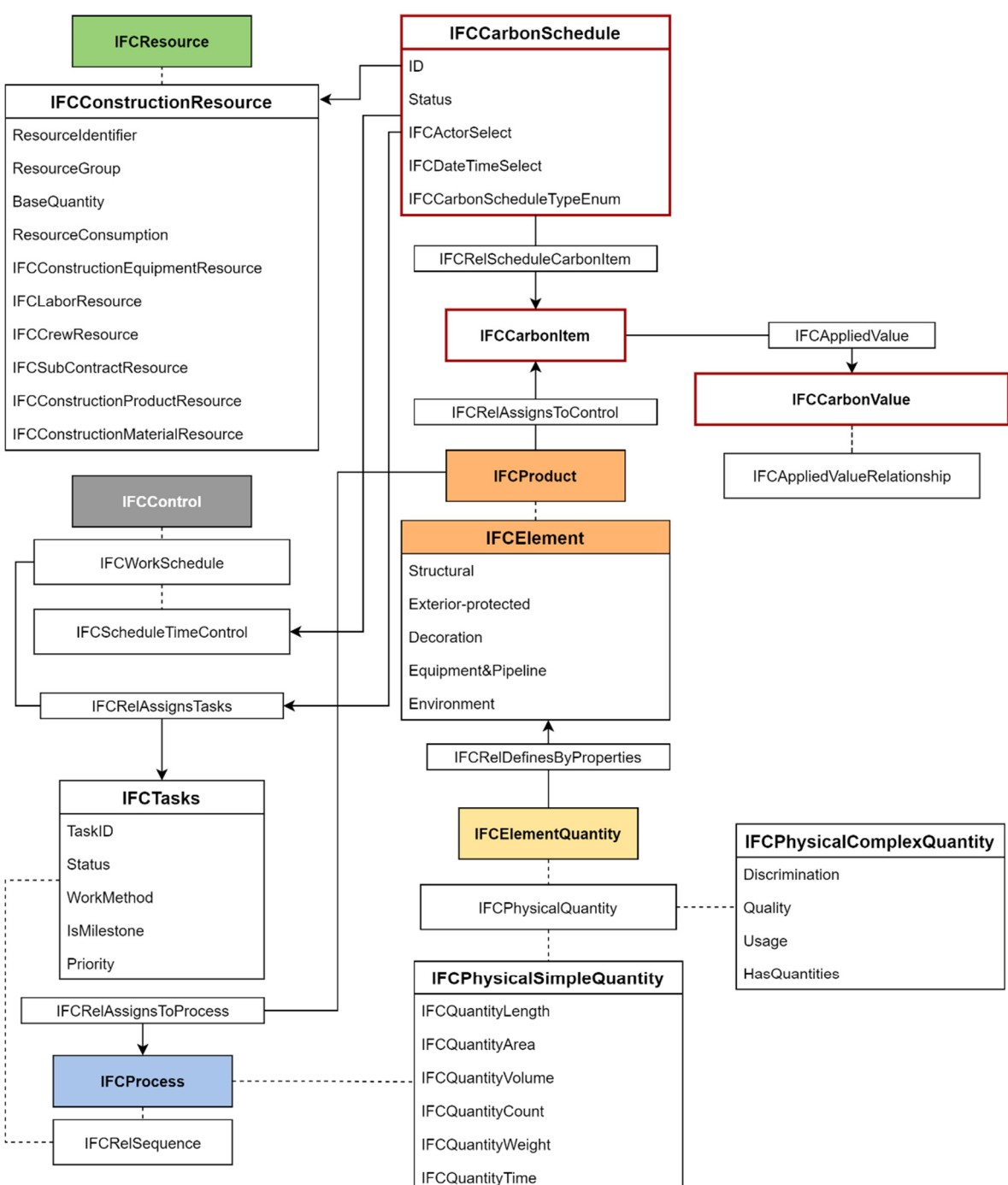

**Figure 5.** IFC data structure and database information conversion.

*3.3. Carbon Emission Database Composition*

(1)　Component Information Table

　　　The component information table is a database that stores the information of building components, as shown in Table 1.

**Table 1.** Component information table.

| Field | Field Type | Field Length |
|---|---|---|
| id | int | 100 |
| name | varchar | 255 |
| type_name | varchar | 255 |
| type_id | int | 100 |
| thing_name | varchar | 255 |
| thing_id | int | 100 |
| thing_number | int | 100 |
| physical | int | 100 |

(2)　Material Information Table

The material information table stores the information of various materials used in the building, as shown in Table 2.

**Table 2.** Material information table.

| Field | Field Type | Field Length |
|---|---|---|
| id | int | 100 |
| name | varchar | 255 |
| type_name | varchar | 255 |
| type_id | int | 100 |
| specs | varchar | 255 |
| carbon_emission_factor_id | int | 100 |
| unit | varchar | 255 |

(3)　Equipment Information Table

The equipment information table stores the information of equipment used in the whole life cycle of the building, as shown in Table 3.

**Table 3.** Equipment information table.

| Field | Field Type | Field Length |
|---|---|---|
| id | int | 100 |
| name | varchar | 255 |
| type_name | varchar | 255 |
| type_id | int | 100 |
| energy_consumption_type | varchar | 255 |
| carbon_emission_factor_id | int | 100 |
| energy_consumption_strength | int | 100 |
| unit | varchar | 255 |

(4)　Personnel Information Table

The personnel information table stores the information of the personnel involved in the whole life cycle of the building, as shown in Table 4.

**Table 4.** Personnel information table.

| Field | Field Type | Field Length |
|---|---|---|
| id | int | 100 |
| name | varchar | 255 |
| gender | varchar | 10 |
| group_id | int | 100 |
| main_skill | varchar | 255 |
| sub_skill | varchar | 255 |
| carbon_emission_factor_id | int | 100 |

(5)   Process Information Table

The process information table stores the relevant information of all the processes in the building life cycle, as shown in Table 5.

**Table 5.** Process information table.

| Field | Field Type | Field Length |
|---|---|---|
| id | int | 100 |
| project_id | int | 100 |
| building_id | int | 100 |
| stage | varchar | 255 |
| processes_type | varchar | 255 |
| components_id | int | 100 |
| components_name | varchar | 255 |
| components_number | int | 100 |
| worker_id | int | 100 |
| worker_number | varchar | 255 |
| worker_working_time | int | 100 |
| equipments_name | varchar | 255 |
| equipments_id | int | 100 |
| equipments_number | int | 100 |
| equipments_working_time | int | 100 |
| specs | varchar | 255 |
| thing_name | varchar | 255 |
| thing_id | int | 100 |
| number | int | 100 |
| unit | varchar | 255 |

(6)   Carbon Emission Factor Table

The carbon emission factor table stores the carbon emission information of various substances that directly and indirectly participate in the building life cycle, as shown in Table 6.

**Table 6.** Carbon emission factor table.

| Field | Field Type | Field Length |
|---|---|---|
| id | int | 100 |
| name | varchar | 255 |
| carbon_emission_type | varchar | 255 |
| carbon_emission_type_id | int | 100 |
| value | int | 100 |
| unit | varchar | 255 |

*3.4. Carbon Emission Calculation Process*

When calculating the total carbon emissions of the building, the program will first determine which stages of the life cycle the project will go through and call the process ID

of each stage of the project from the process table in the database according to the stage information. With the process ID index, the component ID field, process personnel ID field, process equipment ID field, and process material ID field involved in each process can be retrieved from the process information table. Based on these fields, specific information can be located from the corresponding component information table, personnel information table, equipment information table, and material information table in the database (see Table 6 and Figure 6).

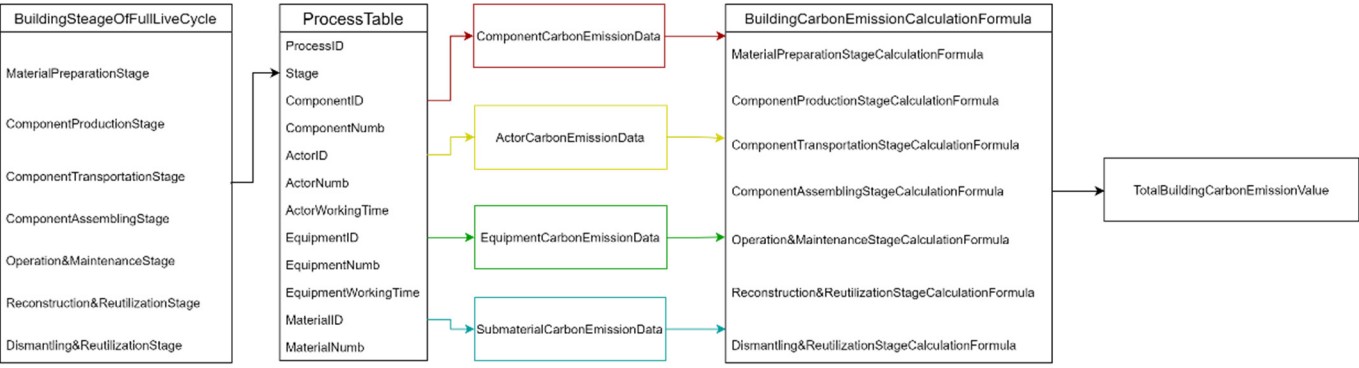

**Figure 6.** Database-based calculation process of carbon emission data.

As mentioned in the first section above, the carbon emission data of each stage of the building consist of component carbon emissions, personnel carbon emissions, equipment carbon emissions, and supporting material carbon emissions. Therefore, the four kinds of carbon emissions of each stage are calculated first and then substituted into the carbon emission calculation formula of each stage.

The calculation process of component carbon emissions (see Figure 7):

(1) Obtain the component ID and component quantity from the process information table;
(2) Obtain the physical parameters (area, volume, dimension, density, mass, etc.), material ID, and material quantity of this type of component from the component information table according to the component ID;
(3) Obtain the carbon emission factor ID of material from the material information table through the corresponding material ID;
(4) Obtain the values and unit of carbon emission factor of material from the carbon emission factor information table through the carbon emission factor ID;
(5) Substitute the obtained component quantity, component physical parameters, material quantity, and material carbon emission factor and unit into $E = \sum_i Qi \times Ci$ for calculation.

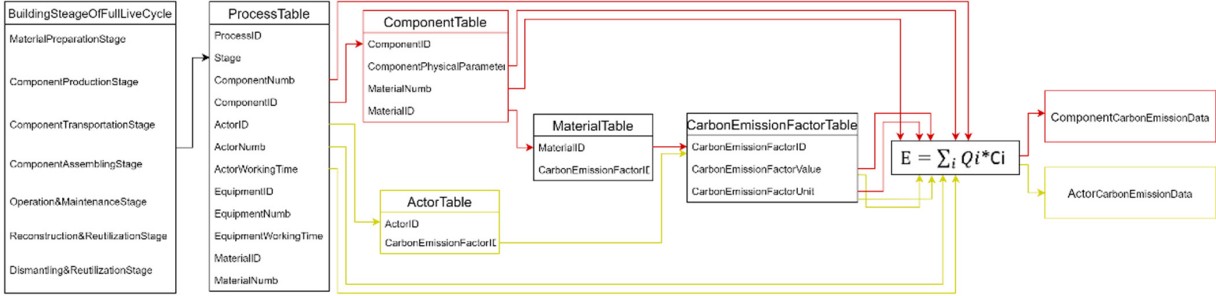

**Figure 7.** Calculation processes of component and personnel carbon emissions.

The calculation process of personnel carbon emissions (see Figure 7):

(1) Obtain the personnel ID, the number of personnel, and the working hours of the personnel from the process information table;

(2) Obtain the carbon emission factor ID of personnel from the personnel information table through the corresponding personnel ID;

(3) Obtain the carbon emission factor value and unit of personnel from the carbon emission factor information table through the carbon emission factor ID;

(4) Substitute the obtained number of personnel, working hours, and personnel carbon emission factor and unit into the corresponding formula $E = \sum_i Qi \times Ci$ for calculation.

The calculation process of equipment carbon emissions (see Figure 8):

(1) Obtain the ID, quantity, and running time of equipment from the process information table;

(2) Obtain the equipment energy consumption carbon emission factor ID and equipment energy consumption intensity of component from the equipment information table through the equipment ID;

(3) Obtain the carbon emission factor value and unit of equipment energy consumption from the carbon emission factor information table through the equipment energy consumption carbon emission factor ID;

(4) Substitute the obtained equipment quantity, equipment energy consumption intensity, equipment running time, and equipment energy consumption carbon emission factor and unit into the corresponding formula $E = \sum_i Qi \times Ci$ for calculation.

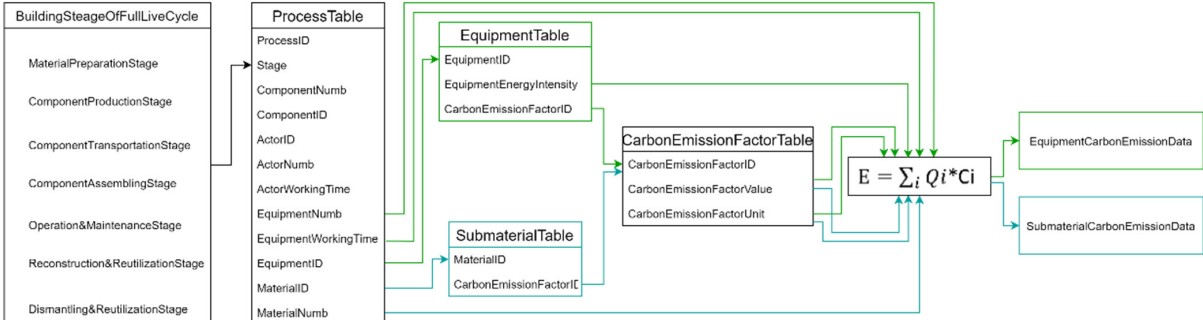

**Figure 8.** Calculation processes of carbon emissions of equipment and supporting materials.

The calculation process of carbon emissions of supporting materials (see Figure 8):

(1) Obtain the ID and quantity of supporting materials from the process information table;

(2) Obtain the carbon emission factor ID of material from the material information table through the corresponding material ID;

(3) Obtain the carbon emission factor values and unit of material from the carbon emission factor information table through the corresponding material carbon emission factor ID;

(4) Substitute the obtained quantity of supporting materials, as well as the value and unit of material carbon emission factor, into $E = \sum_i Qi \times Ci$ for calculation.

After obtaining the four types of carbon emission data in each stage, the carbon emissions of each stage can be obtained by the calculation formula of carbon emissions in each stage of the building life cycle in Section 2. The total carbon emissions of the seven stages are the total carbon emissions of the whole life cycle of the building (see Figure 6).

## 4. Case Study

### 4.1. Case Introduction

C-HOUSE is an entry project of SDC2018. It is a 2-story steel structure residence with a construction area of 183 m². In this paper, the structural components of C-HOUSE are taken as examples to verify the calculation method and process of carbon emissions.

The structural component groups of C-HOUSE consist of a total of 45 steel structural components from 13 categories. The data stored in the building carbon emission database (Table 7) are as follows (for structural components only):

**Table 7.** Component information table.

| Id | Name | Thing_id | Thing_Number | Physical |
|---|---|---|---|---|
| Z2018010101000001 | H welded section steel (long) | 101000001 | 0.36 | 9400 × 200 × 200, 0.046 |
| Z2018010101000002 | H welded section steel (short) | 101000001 | 0.102 | 2900 × 200 × 200, 0.013 |
| Z2018010101000003 | Floor unit a | 101000001 | 0.385 | 2740 × 1090 × 150, 0.049 |
| Z2018010101000004 | Floor unit b | 101000001 | 0.416 | 3240 × 1370 × 150, 0.053 |
| Z2018010101000005 | Floor unit c | 101000001 | 0.385 | 2840 × 1020 × 150, 0.049 |
| Z2018010101000006 | Floor unit d | 101000001 | 0.416 | 3240 × 1150 × 150, 0.053 |
| Z2018010101000007 | Floor unit e | 101000001 | 0.432 | 3440 × 1090 × 150, 0.055 |
| Z2018010101000008 | H welded steel beam | 101000001 | 0.345 | 200 × 200 × 8, 0.044 |
| Z2018010101000009 | Square hollow steel beam | 101000001 | 0.102 | 180 × 180 × 4, 0.013 |
| Z2018010101000010 | Square hollow steel column (short) | 101000001 | 0.165 | 200 × 200 × 2885, 0.021 |
| Z2018010101000011 | Square hollow steel column (long) | 101000001 | 0.33 | 200 × 200 × 5905, 0.042 |
| Z2018010101000012 | Box unit a | 101000001 | 1.028 | 3900 × 200 × 5905, 0.131 |
| Z2018010101000013 | Box unit b | 101000001 | 1.099 | 3200 × 200 × 5905, 0.14 |

The following Table 8 presents material information table.

**Table 8.** Material information table.

| Id | Name | Carbon_Emission_Factor_id | Unit |
|---|---|---|---|
| C0101001 | Hot-rolled steel | C0101005 | Ton |

The following Table 9 presents the equipment information table.

**Table 9.** Equipment information table.

| Id | Name | Energy_Consumption_Type | Carbon_Emission_Factor_id | Energy_Consumption_Strength | Unit |
|---|---|---|---|---|---|
| E01010001 | Plasma cutting machine | Power | C0101001 | 193.6 | kw·h |
| E01010002 | Electric welding machine | Power | C0101001 | 154.6 | kw·h |
| E01010003 | Flatbed truck | Fuel | C0101003 | 20 | L/100 km |
| E01010004 | Truck crane | Fuel | C0101003 | 3.75 | kg/h |
| E01010005 | Platform lift truck | Fuel | C0101002 | 6 | kg/h |

The following Table 10 presents the personnel information table.

**Table 10.** Personnel information table.

| Id | Name | Gender | Carbon_Emission_Factor_id |
|---|---|---|---|
| R01010001 | DZ Zhang | Male | C0101004 |
| R01010002 | YZ Wang | Male | C0101004 |
| R01010003 | H Zhang | Male | C0101004 |
| R01010004 | X Zhang | Male | C0101004 |
| R01010005 | HN Wang | Male | C0101004 |
| R01010006 | S Luo | Male | C0101004 |
| R01010007 | P Liu | Male | C0101004 |
| R01010008 | TR Hua | Male | C0101004 |
| R01010009 | RZ Zhang | Male | C0101004 |
| R01010010 | JC Bu | Male | C0101004 |
| R01010011 | M Zhang | Male | C0101004 |
| R01010012 | CQ Sha | Female | C0101004 |

The following Table 11 presents the process information table.

**Table 11.** Process information table.

| Id | Stage | Components_id | Components_Number | Worker_id | Worker_Number | Worker_Working_Time | Equipments_id | Equipments_Number | Equipments_Working_Time |
|---|---|---|---|---|---|---|---|---|---|
| G0102000001 | Component Production | Z2018010101000001 | 1 | R01010018 | 1 | 0.33 | E01010001 | 1 | 0.33 |
| G0102000002 | Component Production | Z2018010101000001 | 2 | R01010018 | 1 | 0.5 | E01010002 | 1 | 0.5 |
| G0102000003 | Component Production | Z2018010101000002 | 2 | R01010018 | 1 | 0.33 | E01010001 | 1 | 0.33 |
| G0102000004 | Component Production | Z2018010101000003 | 1 | R01010018 | 1 | 1 | E01010001 | 1 | 1 |
| G0102000005 | Component Production | Z2018010101000003 | 1 | R01010018 | 1 | 1 | E01010002 | 1 | 1 |
| G0102000006 | Component Production | Z2018010101000004 | 2 | R01010018 | 1 | 1 | E01010001 | 1 | 1 |
| G0102000007 | Component Production | Z2018010101000004 | 2 | R01010018 | 1 | 1 | E01010002 | 1 | 1 |
| G0102000008 | Component Production | Z2018010101000005 | 3 | R01010020 | 1 | 1 | E01010001 | 1 | 1 |
| G0102000009 | Component Production | Z2018010101000005 | 3 | R01010020 | 1 | 1 | E01010002 | 1 | 1 |
| G0102000010 | Component Production | Z2018010101000006 | 3 | R01010020 | 1 | 1 | E01010001 | 1 | 1 |
| G0102000011 | Component Production | Z2018010101000006 | 3 | R01010020 | 1 | 1 | E01010002 | 1 | 1 |
| G0102000012 | Component Production | Z2018010101000007 | 1 | R01010020 | 1 | 1 | E01010001 | 1 | 1 |
| G0102000013 | Component Production | Z2018010101000007 | 1 | R01010020 | 1 | 1 | E01010002 | 1 | 1 |
| G0102000014 | Component Production | Z2018010101000008 | 2 | R01010020 | 1 | 0.33 | E01010001 | 1 | 0.33 |
| G0102000015 | Component Production | Z2018010101000009 | 17 | R01010019 | 1 | 0.33 | E01010001 | 1 | 0.33 |
| G0102000016 | Component Production | Z2018010101000010 | 6 | R01010018 | 1 | 0.33 | E01010001 | 1 | 0.33 |
| G0102000017 | Component Production | Z2018010101000010 | 6 | R01010021, R01010022 | 2 | 0.5 | E01010002 | 2 | 0.5 |
| G0102000006 | Component Production | Z2018010101000004 | 2 | R01010018 | 1 | 1 | E01010001 | 1 | 1 |
| G0102000007 | Component Production | Z2018010101000004 | 2 | R01010018 | 1 | 1 | E01010002 | 1 | 1 |
| G0102000008 | Component Production | Z2018010101000005 | 3 | R01010020 | 1 | 1 | E01010001 | 1 | 1 |
| G0102000009 | Component Production | Z2018010101000005 | 3 | R01010020 | 1 | 1 | E01010002 | 1 | 1 |
| G0102000010 | Component Production | Z2018010101000006 | 3 | R01010020 | 1 | 1 | E01010001 | 1 | 1 |
| G0102000011 | Component Production | Z2018010101000006 | 3 | R01010020 | 1 | 1 | E01010002 | 1 | 1 |
| G0102000012 | Component Production | Z2018010101000007 | 1 | R01010020 | 1 | 1 | E01010001 | 1 | 1 |
| G0102000018 | Component Production | Z2018010101000011 | 2 | R01010018 | 1 | 0.33 | E01010001 | 1 | 0.33 |

**Table 11.** *Cont.*

| Id | Stage | Components_id | Components_Number | Worker_id | Worker_Number | Worker_Working_Time | Equipments_id | Equipments_Number | Equipments_Working_Time |
|---|---|---|---|---|---|---|---|---|---|
| G0102000019 | Component Production | Z2018010101000011 | 2 | R01010021, R01010022 | 2 | 0.7 | E01010002 | 2 | 0.7 |
| G0102000020 | Component Production | Z2018010101000012 | 2 | R01010021, R01010022 | 2 | 1 | E01010001 | 2 | 1 |
| G0102000021 | Component Production | Z2018010101000012 | 2 | R01010021, R01010022 | 2 | 0.8 | E01010002 | 2 | 0.8 |
| G0102000022 | Component Production | Z2018010101000013 | 2 | R01010021, R01010022 | 2 | 1 | E01010001 | 2 | 1 |
| G0102000023 | Component Production | Z2018010101000013 | 2 | R01010021, R01010022 | 2 | 0.8 | E01010002 | 2 | 0.8 |
| G0103000001 | Component Transportation | Z2018010101000001-Z2018010101000013 | 1 | R01010032, R01010033 | 2 | 8 | E01010003 | 1 | 740.4 |
| G0104000001 | Component Assembly | Z2018010101000013 | 2 | R01010026, R01010027, R01010028, R01010029 | 4 | 0.5 | E01010004 | 1 | 0.2 |
| | | | | | | | E01010005 | 1 | 0.1 |
| G0104000002 | Component Assembly | Z2018010101000012 | 2 | R01010026, R01010027, R01010028, R01010029 | 4 | 0.5 | E01010004 | 1 | 0.2 |
| | | | | | | | E01010005 | 1 | 0.1 |
| G0104000003 | Component Assembly | Z2018010101000011 | 2 | R01010026, R01010027, R01010028, R01010029 | 4 | 0.5 | E01010004 | 1 | 0.2 |
| G0104000004 | Component Assembly | Z2018010101000010 | 2 | R01010026, R01010027, R01010028, R01010029 | 4 | 0.5 | E01010004 | 1 | 0.2 |
| G0104000005 | Component Assembly | Z2018010101000009 | 17 | R01010026, R01010027, R01010028, R01010029 | 4 | 0.3 | E01010004 | 1 | 0.1 |
| | | | | | | | E01010005 | 1 | 0.1 |
| G0104000006 | Component Assembly | Z2018010101000008 | 2 | R01010026, R01010027, R01010028, R01010029 | 4 | 0.4 | E01010004 | 1 | 0.1 |
| | | | | | | | E01010005 | 1 | 0.1 |
| G0104000007 | Component Assembly | Z2018010101000001 | 1 | R01010026, R01010027, R01010028, R01010029 | 4 | 0.5 | E01010004 | 1 | 0.1 |
| G0104000008 | Component Assembly | Z2018010101000002 | 2 | R01010026, R01010027, R01010028, R01010029 | 4 | 0.5 | E01010004 | 1 | 0.1 |

**Table 11.** *Cont.*

| Id | Stage | Components_id | Components_Number | Worker_id | Worker_Number | Worker_Working_Time | Equipments_id | Equipments_Number | Equipments_Working_Time |
|---|---|---|---|---|---|---|---|---|---|
| G0104000009 | Component Assembly | Z2018010101000003 | 1 | R01010026, R01010027, R01010028, R01010029 | 4 | 0.4 | E01010004 | 1 | 0.1 |
| G0104000010 | Component Assembly | Z2018010101000004 | 2 | R01010026, R01010027, R01010028, R01010029 | 4 | 0.4 | E01010004 | 1 | 0.1 |
| G0104000011 | Component Assembly | Z2018010101000005 | 3 | R01010026, R01010027, R01010028, R01010029 | 4 | 0.4 | E01010004 | 1 | 0.1 |
| G0104000012 | Component Assembly | Z2018010101000006 | 3 | R01010026, R01010027, R01010028, R01010029 | 4 | 0.4 | E01010004 | 1 | 0.1 |
| G0104000013 | Component Assembly | Z2018010101000007 | 1 | R01010026, R01010027, R01010028, R01010029 | 4 | 0.4 | E01010004 | 1 | 0.1 |

The following Table 12 presents carbon emission factor table.

**Table 12.** Carbon emission factor table.

| Id | Name | Carbon_Emission_Type | Value | Unit |
|---|---|---|---|---|
| T0101001 | Power (Central China Power Grid) | Power | 0.7035 | $tCO_2 \cdot e/MWh$ |
| T0101002 | Power (East China Power Grid) | Power | 0.8843 | $tCO_2 \cdot e/MWh$ |
| T0101003 | Diesel oil | Fuel | 72.59 | $tCO_2 \cdot e/Tj$ |
| T0101004 | Humankind | Humankind | 0.02 | $tCO_2 \cdot e/d$ |
| T0101005 | Hot-rolled steel | Material | 2.35 | $tCO_2 \cdot e/t$ |

*4.2. Carbon Emission Automatic Computation Based on Database*

Calculation conditions:

The calculation target is the carbon emissions of the structural components provided by the case in the material preparation stage, component production stage, component transport stage, and component assembly stage of the whole life cycle of the building, and the boundaries of the four stages are detailed in Section 2.1.

The carbon emission factors used in the calculation are from Appendix D "Building Material Carbon Emission Factor" of the "Building Carbon Emission Calculation Standard" (GBT 51366-2019) [29]. The equipment information comes from the actual measurement data of the authors' team on the completion process of the case. For complete information, readers can contact the corresponding author of this article.

(1)    Material preparation stage

An example based on the "H welded section steel (long)" member.

Extract the process with the "material preparation stage" field from the process information table, from which retrieve the "H welded section steel (long)" component ID "Z2018010101000001" and the quantity "1"; use the component ID as the index to retrieve the material "hot-rolled carbon steel" ID "C0101001" used for the component and the material quantity "0.36 tons" from the component information table; use the material ID as the index to extract the material carbon emission factor ID "C0101005" from the material information table; use the material carbon emission factor ID as the index to retrieve the carbon emission factor value "2350" and the unit "kg $CO_2$ e/t" of "hot-rolled carbon steel" from the carbon emission factor table. These data are substituted into Equation (3) to obtain the carbon emission of the "H welded section steel (long)" component in the material preparation stage, which is $0.36 \times 2350 \times 1 = 849$ kg $CO_2$. Then, the carbon emission of the structural component group in the material preparation stage is 30,420 kg $CO_2$.

(2)    Component production stage

An example based on the "H welded section steel (long)" member.

Extract the process with the "component production stage" field from the process information table, from which retrieve the personnel ID "R01010018" of "H welded section steel (long)", the number of personnel "1", the working time "0.33", the equipment ID "E01010001", the equipment quantity "1", and the running time of equipment "0.33"; use the personnel ID as the index to retrieve the personnel carbon emission factor ID "C0101004" from the personnel information table; use the equipment ID as the index to retrieve the equipment energy consumption carbon emission factor ID "C0101001", energy consumption intensity "193.6", and unit "$tCO_2$/MWh" from the equipment information table; use the carbon emission factor ID as the index to retrieve the carbon emission factor value "0.7035" and unit "$tCO_2$/MWh" of ID "C0101001" "electricity (Central China Grid)" and the carbon emission factor value "20" and unit "kg$CO_2$/d" of ID "C0101004" "personnel" from the carbon emission factor information table. These data are substituted into Equation (4) to obtain the carbon emission of "H welded steel (long)" component in the component production stage, which is $20/8 \times 0.33 \times 1 \times 1 + 193.6 \times 0.33 \times 0.7035 \times 1 \times 1 = 45.77$ kg $CO_2$. Thus, the carbon emission of the structural component group in the component production stage is 6784.7 kg $CO_2$.

(3)　Component transport stage

An example based on the "H welded section steel (long)" member.

Extract the process with the "component transport stage" field from the process information table, from which retrieve the personnel ID "R01010032, R01010033" corresponding to "H welded section steel (long)", the number of personnel "2", working time "8", equipment ID "E01010003", equipment quantity "1", and equipment transportation distance "740.4"; use the personnel ID as the index to retrieve the personnel carbon emission factor ID "C0101004" from the personnel information table; use the equipment ID as the index to retrieve the equipment energy consumption carbon emission factor ID "C0101003", energy consumption intensity "20", and unit "L/100 km" from the equipment information table; use the carbon emission factor ID as the index to retrieve the carbon emission factor value "72.59" and unit "tCO$_2$/Tj" of ID "C0101003" "diesel" and the carbon emission factor value "20" and unit "kgCO$_2$/d of ID "C0101004" "personnel" from the carbon emission factor information table. The structural component group as a whole is transported by a flatbed truck. These data are substituted into Equation (5) to obtain the carbon emission of structural component groups in the component transport stage, which is 20/8 × 8 × 2 × 1 + 20 × 740.4/100 × 3.3/1000 × 72.59 × 1 × 1 = 75.5 kg CO$_2$.

(4)　Component assembly stage

An example based on the "H welded section steel (long)" member.

Extract the process with the "component assembly stage" field from the process information table, from which retrieve the personnel ID "R01010026, R01010027, R01010028, R01010029" corresponding to "H welded section steel (long)", the number of personnel "4", working time "0.5", equipment ID "E01010004, E01010005", equipment quantity "1, 1", running time of equipment "0.2, 0.1"; use the personnel ID as the index to retrieve the personnel carbon emission factor ID "C0101004" from the personnel information table; use the equipment ID as the index to retrieve the equipment energy consumption carbon emission factor ID "C0101003", energy consumption intensity "3.75", unit "kg/h", "C0101002", energy consumption intensity "6", and unit "kw·h" from the equipment information table; use the carbon emission factor ID as the index to retrieve the carbon emission factor value "72.59" and unit "tCO$_2$/Tj" of ID "C0101003" "diesel", the carbon emission factor value "0.8843" and unit "tCO$_2$/MWh" of ID "C0101002" "electricity (Central China Grid)", and the carbon emission factor value "20" and unit "kgCO$_2$/d" of ID "C0101004" "personnel" from the carbon emission factor information table. The structural component group as a whole is transported by a flatbed truck. Then, these data are substituted into Equation (6) to obtain the carbon emission of the "H welded section steel (long)" component in the component assembly stage, which is 20/8 × 0.5 × 4 × 1 + 3.75 × 0.2 × 0.042 × 72.59 × 1 × 1 + 6 × 0.1 × 0.042 × 72.59 × 1 × 1 = 9.1 kg CO$_2$. Therefore, the carbon emission of the structural component group in the component assembly stage is 251 kg CO$_2$.

It can be seen from the example (Figure 9) that the carbon emission of structural components is the highest in the material preparation stage, accounting for 81.05%, followed by the component production stage, accounting for 18.08%. The carbon emission in the component transport stage and the component assembly stage accounts for a very small proportion, of 0.2% and 0.67%, respectively. Therefore, optimizing the material selection of structural components and the carbon emission in the component production stage can be important means to reduce the carbon emission of components.

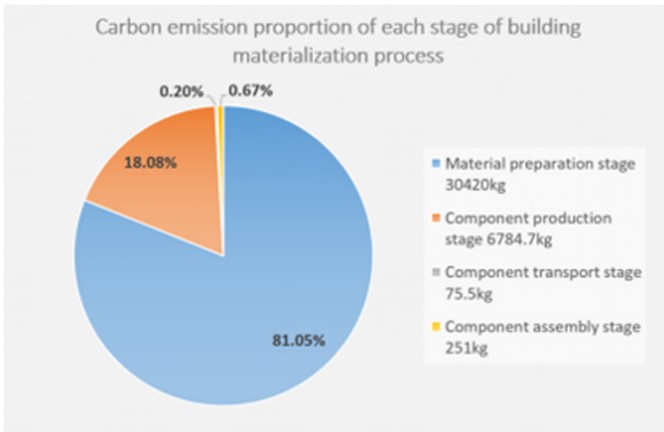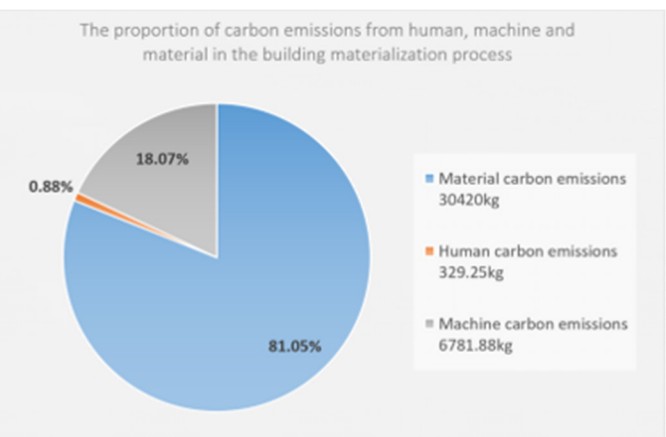

**Figure 9.** Carbon emission proportion of each stage of building materialization process and the proportion of carbon emissions from human, machine and material in the building materialization process.

From the perspective of carbon emission sources of structural components, materials are still the primary source of carbon emissions. Equipment carbon emissions account for a large proportion of 18.07%, and the remaining 0.88% is carbon emissions generated by personnel.

*4.3. Case Comparison and Verification*

In order to verify the dynamic automatic calculation method of carbon emission, this paper will use the carbon emission calculation model and method stipulated by the "Building Carbon Emission Calculation Standard" (GBT 51366-2019) [29] issued by the Ministry of Housing and Urban–Rural Development of China to calculate the carbon emission of the above case.

The definition of the boundary of the whole life cycle of the building in the calculation also comes from the "Building Carbon Emission Calculation Standard" (GBT 51366-2019) [29].

The carbon emission factors used in the calculation are from Appendix D "Building Material Carbon Emission Factor" of the "Building Carbon Emission Calculation Standard" (GBT 51366-2019) [29]; equipment information is from Appendix C "Common Construction Machinery Energy Consumption per Shift" of the "Building Carbon Emission Calculation Standard" (GBT 51366-2019) [29].

(1) Carbon emission of building material production stage = 30,420 kg $CO_2$;
(2) Carbon emission of building material transportation stage = 191.64 kg $CO_2$;
(3) Carbon emission of building construction stage = 9876.11 kg $CO_2$.

**5. Result and Discussion**

*5.1. Results and Analysis*

As can be seen from the above examples, the carbon emissions of material production are exactly the same when the materials are exactly the same. However, after that, whether in the transportation process or the construction process, the results obtained by the two calculation methods were quite different.

According to the calculation method in this paper (Figure 10), the carbon emissions in the transportation stage are only 75.5 kg, but according to the calculation method of GBT 51366-2019 [29], the carbon emissions in the transportation stage are as high as 191.64 kg. The carbon emissions of the construction stage also vary greatly, and the two calculation methods obtain two sets of results, 6784.7 kg and 9876.11 kg, respectively. Even though the boundaries used in this paper are different from GBT 51366-2019 [29], resulting in different calculation ranges between the two, after comparison and sorting, the "material

preparation stage" in this paper corresponds to the "building materials production" stage in the standard. There is no direct corresponding link to the "component production stage", but it can correspond to the "building construction" stage according to the similarity. In this paper, the "logistics transshipment stage" corresponds to the "building materials transportation" stage; the "Component Assembly Phase" corresponds to the "Building Construction" phase. In other words, the carbon emission result of 9876.11 kg in the "building construction" stage in the Standard GBT 51366-2019 [29] corresponds the carbon emission of the "component production stage" + "component assembly stage" in this paper, that is, 6784.7 kg + 251 kg = 7035.7 kg, and there is still a big gap between the two.

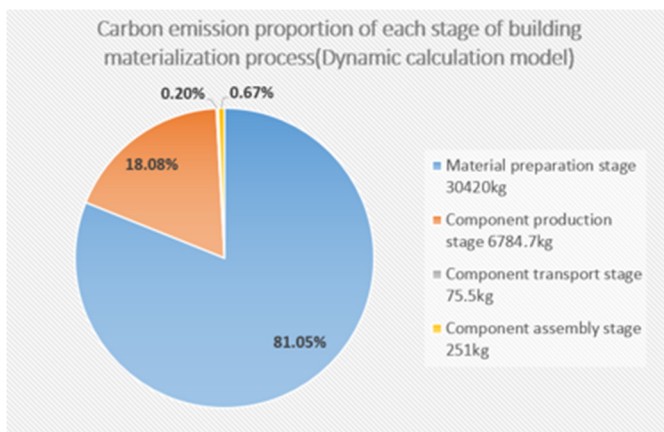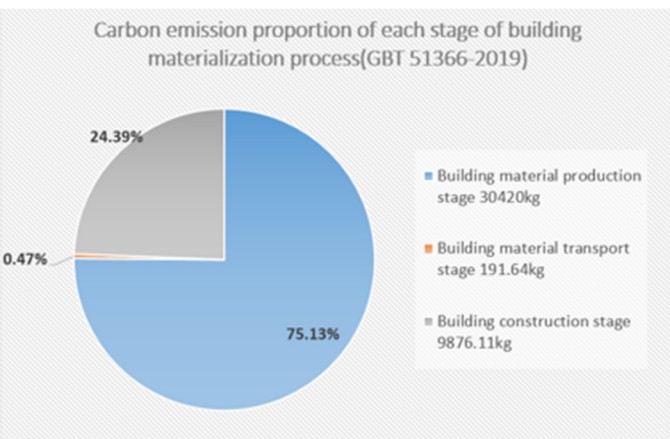

**Figure 10.** Carbon emission proportion of each stage of building materialization process (Dynamic calculation model and GBT 51366-2019) [29].

The results of the two calculation methods, which can be found from the comparison of carbon emission sources, are mainly the difference in the carbon emission results of equipment. The calculation results of this paper show that the carbon emissions of the equipment are 6781.88 kg, and the result of the Standard GBT 51366-2019 [29] is 10,067 kg.

From further study of the calculation formula of equipment carbon emissions of the two methods, it can be found that the two are basically the same in the statistics of energy consumption of equipment, and the biggest difference is that the calculation method in this paper uses the real operation time of the equipment recorded by the project, while the Standard GBT 51366-2019 [29] uses the time quota corresponding to the engineering quantity (Figure 11). Since the quota cannot be adjusted in real time according to the building type, construction technology, worker proficiency, and construction site environmental variables, the results of this comparative analysis are significantly higher than those of the actual situation, and the deviation rate is far beyond the allowable deviation range. Therefore, when estimating the carbon emissions of the project in the early stage of the building plan, the calculation method of the quota cannot obtain sufficiently accurate estimation results, which is unfavorable to the optimization and adjustment of the building plan.

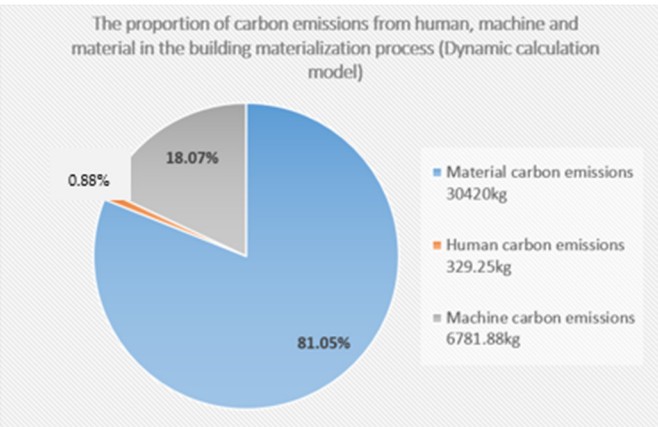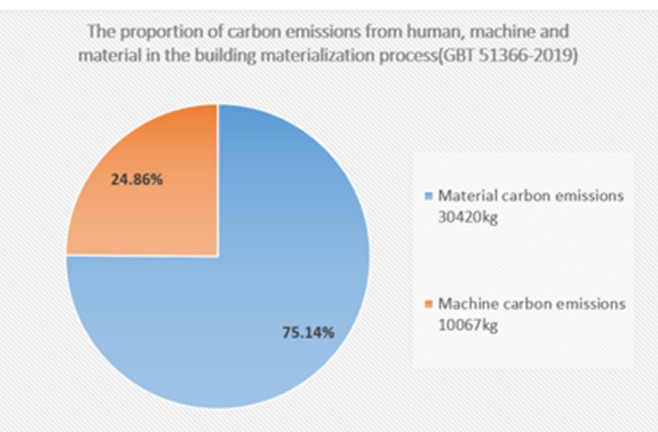

**Figure 11.** The proportion of carbon emissions from human, machine and material in the building materialization process (Dynamic calculation model and GBT 51366-2019) [29].

*5.2. Static and Dynamic Carbon Emission Calculation Models*

In view of the above two computational models based on different temporal information sources, this paper argues that the two computational models can be defined as a dynamic model system and static model system.

The dynamic model system, regards time, a fourth-dimensional engineering variable, as a global quantitative parameter. Time information no longer only plays a role in node identification but becomes an inseparable part of the engineering entity and interacts with and affects the rest of the information of the model system. The advantage of such a model system is that it reflects the real engineering situation. In addition, the optimization of the time schedule by the project management will be reflected in the overall model, and the model can be used to debug various project schedule schemes to obtain a more optimal project schedule design. At the same time, the carbon emission information greatly influenced by the time variable will be more realistic and accurate. The disadvantage lies in the introduction of a variable that can affect the whole model, which will lead to more complex relations between entities in the model system. Furthermore, the system is upgraded from three-dimensional to four-dimensional, which will greatly increase the information density.

A static model system is a model system which is based on a material entity and deduces the dynamic information such as the project time node through the project quota. In other words, all engineering information in this system is determined by the three-dimensional physical parameters of the model, and engineering information of other dimensions is not involved in the quantitative calculation but only serves as reference indexes. The advantage of such a model system is that the data are based on the empirical quota accumulated in long-term engineering practice, and the source is reliable and authoritative. Nevertheless, it cannot reflect the real situation of the project, and it cannot count or evaluate the dynamic variables in the project.

By comparing the two model systems, it is concluded that the dynamic model system is superior to the static one in simulation and automatic calculation.

**6. Conclusions**

This paper proposes a method for calculating building carbon emissions based on a BIM general data framework, which uses BIM technology to automatically extract building information and match it with a carbon emission database, realizing the rapid calculation of carbon emissions in the whole life cycle of buildings. This paper also constructs a BIM general data framework based on the IFC standard, which converts BIMs generated by different software into a unified data format and defines the carbon emission calculation rules for each life cycle stage. Finally, this paper takes a prefabricated residential building as an example to verify the feasibility and effectiveness of this method and compares it

with the traditional method. The results show that this method can accurately calculate the carbon emissions of each life cycle stage and each component type and reflect the differences in carbon emission impacts of different design schemes. This method provides a new idea and tool for building carbon emission management. Future research will further improve the practicality and accuracy of this method by combining automatic acquisition of project progress and intelligent perception of the site situation.

**Author Contributions:** Conceptualization, R.Z.; methodology, R.Z.; data curation, H.Y.; writing—original draft, R.Z.; writing—review & editing, S.H.; supervision, H.Z. All authors have read and agreed to the published version of the manuscript.

**Funding:** This study was funded by the Ministry of Science and Technology and National Key R&D Program of China "Fourteenth Five-Year Plan" project "High-quality Green Building Design Method and Intelligent Collaborative Platform"/2022YFC3803804.

**Institutional Review Board Statement:** Not applicable.

**Informed Consent Statement:** Not applicable.

**Data Availability Statement:** Not applicable.

**Conflicts of Interest:** The authors declare no conflict of interest.

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
