# Peer review of "Research on Database Construction and Calculation of Building Carbon Emissions Based on BIM General Data Framework"

_sustainability, doi:10.3390/su151310256_

Round 1

Reviewer 1 Report

The manuscript "Research on database construction and calculation of building carbon emissions based on BIM general data framework" proposes a method for calculating the life cycle carbon emission (LCCE) of prefabricated buildings based on building information modeling (BIM) technology. The method identifies seven stages of the prefabricated building life cycle: material preparation, component production, component transport, component assembly, operation and maintenance, renovation and reuse, and demolition and reuse. This paper presents an interesting approach and in terms of data, this paper is indeed very rich. Also, I enjoyed the part related to the case study. I recommend this manuscript for publication after following mandatory revisions.

1.     This paper needs a major revision on the introduction. To me, it is more a report of a modelling approach without much explanation from the background. The introduction on recent developments in carbon-related policies and challenges needs an improvement. The following articles on the issue are highly recommended to be used, as they contain relevant discussions on the topic:

-        Guo, B., Feng, Y., & Hu, F. (2023). Have carbon emission trading pilot policy improved urban innovation capacity? Evidence from a quasi-natural experiment in China. Environmental Science and Pollution Research. doi: 10.1007/s11356-023-25699-x

-        Feng Hu, Liping Qiu, Yang Xiang, Shaobin Wei, Han Sun, Hao Hu, Xiayan Weng, Lidan Mao, Ming Zeng, (2023). Spatial network and driving factors of low-carbon patent applications in China from a public health perspective. Frontiers in Public Health. doi: 10.3389/fpubh.2023.1121860

-        Guo, B., Wang, Y., Zhou, H., & Hu, F. (2022). Can environmental tax reform promote carbon abatement of resource-based cities? Evidence from a quasi-natural experiment in China. Environmental science and pollution research. doi: 10.1007/s11356-022-23669-3

2.     Most of the texts in tables are hardly readable (for instance Figure 6). Please consider replacing them with higher quality images.

3.     What does the table in Line 473 add to the manuscript?

4.     Can you please comment on the errors of measurements? There is hardly anything on the error calculations.  

5.     Can you please comment on the weight of influencing parameters? I mean which part has the highest contribution?

6.     Does your model consider the carbon footprint in raw materials? Please explain.

Author Response

Point 1:     This paper needs a major revision on the introduction. To me, it is more a report of a modelling approach without much explanation from the background. The introduction on recent developments in carbon-related policies and challenges needs an improvement. The following articles on the issue are highly recommended to be used, as they contain relevant discussions on the topic:

-        Guo, B., Feng, Y., & Hu, F. (2023). Have carbon emission trading pilot policy improved urban innovation capacity? Evidence from a quasi-natural experiment in China. Environmental Science and Pollution Research. doi: 10.1007/s11356-023-25699-x

-        Feng Hu, Liping Qiu, Yang Xiang, Shaobin Wei, Han Sun, Hao Hu, Xiayan Weng, Lidan Mao, Ming Zeng, (2023). Spatial network and driving factors of low-carbon patent applications in China from a public health perspective. Frontiers in Public Health. doi: 10.3389/fpubh.2023.1121860

-        Guo, B., Wang, Y., Zhou, H., & Hu, F. (2022). Can environmental tax reform promote carbon abatement of resource-based cities? Evidence from a quasi-natural experiment in China. Environmental science and pollution research. doi: 10.1007/s11356-022-23669-3

Response 1: Thank you for your suggestions. I have added some background and policy explanations on carbon emissions in the introduction section. I also appreciate the three articles you provided. They were very helpful and I have included them as part of my references.

Point 2:    Most of the texts in tables are hardly readable (for instance Figure 6). Please consider replacing them with higher quality images.

Response 2: Thank you for your reminder. I have remade the figures in the article in high-definition format.

Point 3:     What does the table in Line 473 add to the manuscript?

Response 3: The table in line 473 of the original article records the process information of the case, which is the core component of the calculation method in this paper. The table records the information involved in each step of the building life cycle, including the people, machines, materials and time information involved. Based on the information recorded in this table, the background program can correctly call the data in other tables and realize the automatic calculation of carbon emissions.

Point 4:     Can you please comment on the errors of measurements? There is hardly anything on the error calculations.  

Response 4: Thank you for your suggestions. The article has added a case comparison and verification section, which uses the building carbon emission calculation method in the Chinese "Building Carbon Emission Calculation Standard" (GBT 51366-2019) to compare and verify the case. And based on the calculation results, the data differences caused by the two calculation methods are analyzed.

Point 5:   Can you please comment on the weight of influencing parameters? I mean which part has the highest contribution?

Response 5: The discussion section of the article has added an analysis of the carbon emission weights, which shows that the material carbon emission still accounts for the highest proportion of the carbon emission in the whole life cycle of the building, and the calculation of the whole process mechanical carbon emission has the greatest impact on the calculation accuracy of the carbon emission in the construction stage of the building. The data of the actual running time of the machinery has the greatest influence on the prediction of the building carbon emission.

Point 6:     Does your model consider the carbon footprint in raw materials? Please explain.

Response 6: Thank you for your question. The part of the calculation model in this paper about the material carbon footprint is included in the calculation of the material preparation stage. The material carbon emission factor value is derived from Appendix D of the Chinese "Building Carbon Emission Calculation Standard" (GBT 51366-2019) Building Material Carbon Emission Factor.

Reviewer 2 Report

  The article "Research on database construction and calculation of building carbon emissions based on BIM general data framework" introduces a carbon emission calculation modeling in BIM. The article does not effectively introduce the research topic and does not pose the research question and gap clearly. The article does not highlight the significance and novelty of the study in clear and concise manner. The important quantitate information of references is missing and thus lacks the coherence in general. Moreover, the authors state that a lot of work has been already done of BIM based carbon emission calculation. However, at the end of the introduction section it is simply mentioned that there is some room for improvement which makes the novelty of the article is questionable. Methodology section is also weak. The authors without any reference or basis divide the carbon emission calculations in 7 stages and give the mathematical equations. There is no validation or any sound backing of the methodology presented in the paper. After that the authors discuss how to enter the data and used to run it in BIM which looks like a user guide rather than a research publication. A case study is performed for a two-story building and the results are presented. There is no comparison with any other carbon emission standards or calculation which leaves a question mark on the quality and accuracy of results. The results and discussion section is too short and does not have any substance.

Therefore, this article is recommended for rejection. 

English language is ok.

Author Response

Point 1:   The article does not effectively introduce the research topic and does not pose the research question and gap clearly.

Response 1: Thank you for your suggestions on the introduction section of this paper. I have rewritten the introduction section, focusing on adding the introduction of the research topic, and emphasizing the difference between the research method of this paper and the existing research.

Point 2:   The article does not highlight the significance and novelty of the study in clear and concise manner.

Response 2: Thank you for your correction. I have rewritten the introduction section and tried to describe the value and innovation of the research in a clearer way.

Point 3:   The important quantitate information of references is missing and thus lacks the coherence in general.

Response 3: According to your opinion, I have supplemented the research background and reorganized the clues of the literature to enhance the coherence.

Point 4:   Moreover, the authors state that a lot of work has been already done of BIM based carbon emission calculation. However, at the end of the introduction section it is simply mentioned that there is some room for improvement which makes the novelty of the article is questionable.

Response 4: Thank you for your reminder. In order to clarify the existing problems and explain the problems that the article wants to solve, the end of the introduction section has been modified, emphasizing that the existing carbon emission calculation methods lack the ability to intuitively provide architects with the prediction of the whole life cycle carbon emission of the building scheme at the scheme stage, and also cannot provide the impact weight of various types of carbon emission at the design stage, providing data basis for scientifically reducing the total carbon emission of the project. The research of this paper on the automatic calculation method of building carbon emission based on BIM is to supplement this part of the content, and provide theoretical basis for the construction of an automated platform that integrates building model modeling, calculation and optimization.

Point 5:    Methodology section is also weak. The authors without any reference or basis divide the carbon emission calculations in 7 stages and give the mathematical equations.

Response 5: Thank you for your correction. The division of the seven stages of the whole life cycle of the building is based on the European standard (BS EN15978:2011), the Chinese standard (GBT 51366-2019) and the reference literature mentioned in the introduction. After summarizing the above information, and according to the requirements of most cities in China for building industrialization, and the initiative for building reuse, it is extended to seven stages. The original text did not explain this derivation process, and now it has supplemented this content.

Point 6:   There is no validation or any sound backing of the methodology presented in the paper.

Response 6: Thank you for your criticism. I have added the theoretical basis for the division of the whole life cycle of the building in section 2.1, and supplemented the research support for the IFC data structure in section 2.3. I hope to improve the credibility of the method section of this paper.

Point 7:  After that the authors discuss how to enter the data and used to run it in BIM which looks like a user guide rather than a research publication. 

Response 7: The logic of the content of Section 3 Establishing dynamic calculation method based on carbon emission database has been adjusted, and 3.1 "The lack of recording of real-time information of engineering in IFC data structure" has been added. The logical structure of Section 3 is modified as follows: after establishing the building carbon emission information structure based on IFC data structure in Section 2, Section 3.1 considers that the storage of real-time information of engineering cannot be realized under the existing IFC data structure, Section 3.2 decides to introduce database system to supplement the missing information, Section 3.3 introduces the specific composition form of the added database, and Section 3.4 shows the process of retrieving various types of building information when performing automatic calculation based on the above data structure. I hope the current changes are more in line with academic expression.

Point 8:   A case study is performed for a two-story building and the results are presented. There is no comparison with any other carbon emission standards or calculation which leaves a question mark on the quality and accuracy of results.

Response 8: Thank you for your suggestions. I have added section 4.3 "Case Comparison and Verification" to the case analysis chapter, which uses the building carbon emission calculation method in the Chinese "Building Carbon Emission Calculation Standard" (GBT 51366-2019) to compare and verify the case. The verification proves that using time information close to the actual situation of the project has a large difference from using quota information based on engineering quantity in carbon emission calculation, and the former is more accurate. The data used in the article case are real data recorded on the project site, which are inconvenient to append due to the large amount of data, and only one page of record samples as follow. For complete information, please contact the corresponding author of this paper.

Point 9:   The results and discussion section is too short and does not have any substance.

Response 9: This paper has rewritten section 5 "Result and discussion". 5.1 "Results and analysis", focusing on analyzing the reasons for the difference in calculation results between the new method proposed in this paper and the quota method used for comparison and verification, and concluding that the reason for such difference is the different value of mechanical running time. 5.2 "Static and dynamic carbon emission calculation models", comparing and analyzing the calculation models used by the two methods, and concluding that the dynamic model system has more advantages in achieving simulation automatic calculation.

Round 2

Reviewer 1 Report

Dear authors, thanks for detailed and extensive revision. I suggest this paper for publication. 

Reviewer 2 Report

The authors have significantly improved the manuscript and addressed all my reservations. This article can now be recommended for acceptance.

Minor language checks are required.